# Familiarization with Mixed Reality for Individuals with Autism Spectrum Disorder: An Eye Tracking Study

**DOI:** 10.3390/s23146304

**Published:** 2023-07-11

**Authors:** Maxime Leharanger, Eder Alejandro Rodriguez Martinez, Olivier Balédent, Luc Vandromme

**Affiliations:** 1UR 7516 Laboratory CHIMERE, University of Picardie Jules Verne, 80000 Amiens, France; eder.rodriguez@u-picardie.fr (E.A.R.M.); olivier.baledent@chu-amiens.fr (O.B.); luc.vandromme@u-picardie.fr (L.V.); 2Institut Faire Face, CHU Amiens, 80054 Amiens, France

**Keywords:** mixed reality, ASD, autism, eye tracking, familiarization, training, learning, visual attention

## Abstract

Mixed Reality (MR) technology is experiencing significant growth in the industrial and healthcare sectors. The headset HoloLens 2 displays virtual objects (in the form of holograms) in the user’s environment in real-time. Individuals with Autism Spectrum Disorder (ASD) exhibit, according to the DSM-5, persistent deficits in communication and social interaction, as well as a different sensitivity compared to neurotypical (NT) individuals. This study aims to propose a method for familiarizing eleven individuals with severe ASD with the HoloLens 2 headset and the use of MR technology through a tutorial. The secondary objective is to obtain quantitative learning indicators in MR, such as execution speed and eye tracking (ET), by comparing individuals with ASD to neurotypical individuals. We observed that 81.81% of individuals with ASD successfully familiarized themselves with MR after several sessions. Furthermore, the visual activity of individuals with ASD did not differ from that of neurotypical individuals when they successfully familiarized themselves. This study thus offers new perspectives on skill acquisition indicators useful for supporting neurodevelopmental disorders. It contributes to a better understanding of the neural mechanisms underlying learning in MR for individuals with ASD.

## 1. Introduction

Autism Spectrum Disorder (ASD) is a multifactorial neurodevelopmental disorder mainly of genetic origin, and its prevalence is about 1%. The DSM-5 (Diagnostic and Statistical Manual of mental disorders) [1] considers it a single spectrum characterized by two criteria: a deficit in communication and social interactions and restricted and repetitive patterns of behavior and interests, including hypersensitivity and/or hyposensitivity to certain stimuli. However, this deficiency in expressing their emotions, particularly in school, where children accumulate delays that are never caught up. Moreover, no specific method exists to increase autonomy in people with ASD. Information and Communication Technologies (ICT) have been identified as having significant potential to improve the lives of individuals with ASD by providing personalized and adapted interventions and therapeutic approaches to their specific needs [2]. Several areas of ICT application for ASD have been explored, including education, therapy, and vocational training [3].


**Educational Technologies**


ICT can be used in education to develop personalized and adaptive teaching programs for individuals with ASD [4]. Educational applications, serious games, and online learning platforms can be designed to facilitate the acquisition of cognitive, social, and behavioral skills [5]. Video games can offer immersive and interactive environments that encourage individuals with ASD to explore and solve problems [6]. These video games, available in various formats such as tablets, laptops, and virtual reality (VR) or mixed reality (MR) headsets, provide immersive, interactive, and flexible learning environments that can be adapted to the individual needs and preferences of people with ASD [7]. These tools can provide individualized learning experiences tailored to the specific needs of each individual [8].

Additionally, Wearable Assistive Technologies (WAT) are increasingly being used to support individuals with ASD in their daily lives. These devices can monitor the behaviors, emotions, and physiological activities of individuals with ASD, providing valuable information for caregivers, therapists, and researchers. For example, physical activity tracking devices can help promote exercise and physical activity among people with ASD, improving their overall health and well-being [9].

Studies have also examined the effectiveness of robots and virtual agents in providing social interventions for children with ASD, with promising results in improving communication and social engagement and reducing repetitive behaviors [10,11]. Research has shown that interactions with robots can enhance engagement and motivation for individuals with ASD, particularly when faced with social challenges [12].

Studies have shown that children with ASD can benefit from using VR to improve their social and emotional skills [13] and mobile applications to enhance their communication skills [14,15]. The MR headset has also been explored to support individuals with neurodevelopmental disorders by offering immersive and interactive learning experiences [16]. VR and Augmented Reality (AR) interventions are increasingly used to help individuals with ASD develop social and emotional skills [2]. These technologies allow individuals to immerse themselves in controlled virtual environments where they can practice social interactions without the challenges and distractions of the real world. For example, HoloLearn is a mixed-reality platform aimed at improving individuals' social and emotional skills with neurodevelopmental disorders [16].


**Therapies and Interventions**


ICT can also support therapies and interventions for individuals with ASD. For example, computer-assisted therapy applications and telehealth platforms allow therapists and healthcare professionals to provide remote, real-time interventions for individuals with ASD who may have difficulty accessing in-person services [15]. Furthermore, assistive technologies, such as alternative and augmentative communication (AAC) devices, can help individuals with ASD improve their communication and social interaction skills [17]. ICT can also facilitate vocational training and employment for individuals with ASD. Specific applications and programs have been developed to help individuals with ASD acquire and strengthen the skills needed to succeed in the job market. These tools can include simulations of professional situations, serious games, and training programs based on VR or mixed reality, which allow individuals with ASD to practice and adapt to a work environment in a safe and controlled setting.


**Mixed Reality**


The HoloLens 2 headset uses Microsoft's mixed reality (MR) technology [18]. Under development in the industrial and healthcare sectors, this technology allows for the real-time overlay of 3D virtual objects in the user’s environment [19,20]. Thus, MR provides additional information to help users perform tasks while interacting with their real environment. MR lies on a continuum between augmented reality (AR) and virtual reality (VR). The immersion offered by MR increases user engagement and motivation, which is crucial for individuals with ASD, who may have attention deficits or sensory sensitivities that influence their desire to learn [1]. AR, which overlays virtual objects onto the real environment using digital devices such as tablets, smartphones, or glasses, offers limited interactivity and immersion compared to VR [21]. In contrast, VR offers total virtual immersion and has been used to improve social skills in individuals with ASD [13]. However, some studies highlight the difficulties in generalizing skills acquired through new technologies in real life [22]. MR allows for training with virtual objects while interacting with real stimuli, thus overcoming the limitations of VR and AR.

Moreover, some studies report user fatigue due to the insufficient development of digital environments as rich as the real world [23]. Furthermore, prolonged use of VR can cause motion sickness [24], reducing the impact of total immersion and usage time. The apprehension of motion sickness when using VR poses a risk factor for continued learning. The HoloLens 2 glasses overcome the limitations of VR and AR by offering a balance between immersion and interactivity with the real world [25].


**Eye-tracking and Mixed Reality**


ASD is a neurodevelopmental disorder that affects social interactions, communication, and behaviors and may be related to anomalies in eye contact [26]. Eye-tracking (ET) has revealed difficulties in capturing social information in people with autism-related to atypical attention processes [27,28]. These skills are crucial for a child’s cognitive development, particularly for joint attention [29]. Over the decades, ET has become an indispensable tool for studying visual activity [30]. In parallel, MR technologies have emerged, combining elements of VR and AR to create immersive experiences. The extended reality (XR) devices incorporating ET are the HTC Vive Pro Eye, Pico Neo 2 Eye, and Varjo XR-3. These devices combine the advantages of VR, AR, and MR with ET to provide even more immersive and personalized experiences. Using ET in these devices can help better understand attention mechanisms, facilitate navigation and interactions, and optimize graphical performance by adapting resolution based on gaze point. An example of this synergy between MR and ET is found in the Microsoft HoloLens 2 device. This technology combines MR with ET to provide accurate information on gaze position, allowing a better understanding of attention mechanisms in individuals with ASD [31]. The source code of the ARRET eye-tracking toolkit developed by the University of Kaiserslautern is freely provided and must be adapted to the experimental design “https://github.com/AR-Eye-Tracking-Toolkit (accessed on 3 June 2023)”. Integrating ET into MR devices allows the collection of valuable data for studying neurodevelopmental disorders, such as ASD. However, methodological challenges related to data analysis and experimental design must be carefully addressed to exploit these innovative tools in neuroscience research fully. For example, it is necessary to determine the gaze classifiers appropriate for the experiment and to consider the experimental design to adjust the visual angle thresholds and the temporal window [32].

Furthermore, it is important to determine, among the many dependent variables (number, duration of fixations or saccades) and the numerous areas of interest (holograms), which ones are indicative of learning during the use of MR [33]. In summary, the combination of MR technology and ET, as in the case of Microsoft HoloLens 2, opens up new perspectives for studying neurodevelopmental disorders such as ASD. Integrating these two technologies allows for immersive and interactive experiences while collecting accurate data on visual activity and participants’ attention processes. By incorporating ET into MR devices, researchers can develop more advanced methodologies for studying neurodevelopmental disorders like ASD.


**Familiarization**


Our primary goal is to familiarize individuals with ASD with wearing the headset and using MR technology. Familiarization with new technologies or situations for individuals with ASD can be facilitated by specific methods tailored to their needs. A study conducted by [34] demonstrated the effectiveness of a habituation program in reducing stress among autistic children during MRI exams. This program included simulation sessions (2 with a mock MRI) of the exam with MRI noises and explanatory videos, allowing participants to understand better and adapt to the MRI environment. In a medical context, [35] developed a gradual method to help individuals with autism become accustomed to injections and blood draws. The method involved gradual exposure to needle-related stimuli, starting with play activities and progressing towards imitating the actual situation. This approach increased needle tolerance and reduced problematic behaviors during medical procedures. They conducted a study involving 50 sessions and concluded that the complexity of the task might warrant additional sessions. In the technology field, an example of a virtual reality (VR) application to help individuals with ASD enhance their social cognition skills is presented by [13]. They developed a VR training program for young adults with high-functioning autism. The program simulated real-life scenarios requiring understanding others’ perspectives and recognizing emotions, thus creating an immersive environment for social cognition training. The participants attended ten sessions of this innovative training over five weeks. The results showed significant improvements in emotion recognition and theory-of-mind abilities among participants. This highlights the potential of VR as an effective tool for social cognition training in autism, providing an engaging medium for individuals with ASD to adapt and familiarize themselves with new situations.


**Assessing Changes in Behavior and Attention during Learning**


Considering these familiarization methods, we aim to explore how MR and HoloLens 2 could facilitate learning in individuals with ASD. Furthermore, few studies have quantified changes in behavior during learning in individuals with ASD. HoloLens 2 serves as a relevant device for gathering these learning indicators. One of the learning principles is learning through practice, which means repeating a task. The effects of the practice are characterized by an increase in task execution speed and a reduction in the number of errors during trials [36]. Thus, our second objective is to collect learning indicators, such as task execution speed or changes in attention inferred through gaze over time. Quantifying a change in gaze is crucial data, especially for individuals with ASD with a unique gaze pattern [37]. Understanding and analyzing changes in gaze during learning can help tailor and personalize interventions for individuals with ASD, considering their specific needs and attentional skills. Several studies have highlighted the importance of gaze analysis in evaluating and monitoring interventions for individuals with ASD [27,38]. For example, using eye-tracking to analyze the visual preferences of children with autism revealed differences in how they explore and process social stimuli, such as faces [39]. By integrating MR technology and ET in the HoloLens 2, we can collect valuable data on changes in gaze and attention during learning in individuals with ASD. This information can be used to tailor interventions, considering each individual’s strengths and challenges, and providing personalized support to maximize learning outcomes [40].

Additionally, analyzing gaze data collected using HoloLens 2 could help better understand the underlying learning mechanisms in individuals with ASD. For example, we could examine how changes in gaze are related to error reduction and improved performance over time, as well as the generalization of acquired skills in other everyday contexts [41]. In summary, our study aims to explore the use of MR and ET technology in HoloLens 2 to familiarize individuals with ASD with new technologies and situations and quantify changes in behavior and attention during learning. The results of this research could contribute to improving learning methods and interventions for individuals with ASD, providing valuable insights into attentional processes and adaptation to new environments.

## 2. Materials and Methods

### 2.1. Participants 

In this cross-sectional quantitative study, the control group consists of fifty-nine neurotypical individuals (mean age: 21.8 years; standard deviation: 4.24 years), comprising fifty-one females and nine males. Participants were recruited through word of mouth and advertisements. For the NT group (control group of “healthy volunteers”), the inclusion criteria are as follows:Participants must be older than ten years of age;Participants cannot have a diagnosis of ASD;Participants must not exhibit psychiatric disorders such as attention deficit disorder with or without hyperactivity, depression, bipolar disorders, or schizophrenia.Participants must not have a neurological history that includes conditions like epilepsy or cerebrovascular accidents;Participants must provide their informed consent verbally and in writing after receiving comprehensive information about the study.

The group of individuals with ASD consists of 11 people (mean age: 24.1 years; standard deviation: 7.5 years), including nine men and two women. These participants were recruited through research partnerships in two medical-social structures of the ADAPEI 80 association. For the groups of individuals with autism spectrum disorder, the inclusion criteria are as follows:Participants must be older than ten years of age;Participants had at least one ASD diagnosis;Participants must provide their informed consent verbally and in writing after receiving comprehensive information about the study.

The demographic data and clinical measures on individuals with ASD are presented in Table 1.

### 2.2. Methodology 

This study aims to evaluate the effectiveness of a familiarization procedure to assist individuals with ASD in using an MR headset to perform learning tasks and manipulate holograms.

#### 2.2.1. Familiarization with the MR Headset 

Familiarization with the MR headset is crucial for people with ASD to ensure their comfort and adaptation to this technology, especially when hypersensitive to certain stimuli [1]. The familiarization steps are inspired by the basic principles of the learning method. They include customization, repetition through practice with progressively more intensive levels of difficulty, and the use of positive reinforcement. We are inspired by the study by Nordahl and his team, which proposes several progressive stages to familiarize people with ASD to perform an MRI examination by cutting out each key stage of the examination [34]. In the same way, we propose progressive stages. The objective of our study is that individuals can easily use the headset at the end of familiarization. These MR helmet familiarization steps can be divided into two main parts: familiarization with wearing the helmet itself and familiarization with the MR. 

#### 2.2.2. Familiarization with the Headset 

This first part aims to desensitize participants to the various factors related to the headset, such as its weight, how it fits in the user’s field of vision, and the temperature associated with the operation of the processor located at the back of the headset. For this purpose, we use a 3D-printed dummy headset called HoloMax (Figure 1), which weighs the same as the real headset and is adaptable to all head shapes. Participants go through various stages, from simply being shown the headset to wearing it for several minutes.

#### 2.2.3. Familiarization with Mixed Reality 

The second part focuses on understanding basic movements for interacting with holograms when using the headset. We first ask volunteers to observe holograms in the MRTK2 application (Mixed Reality Tool Kit 2 [18]), allowing them to experiment with examples of interaction with holograms, such as a piano, a sphere, push buttons, etc. If they agree to observe the holograms and attempt to interact with them, they can move on to the next stage. People with ASD may have particular difficulties during ocular calibration, in particular, because of the associated atypical visual attention [26,27,37,38]. These characteristics can lead to calibration errors and affect the accuracy of the data collected. Prior familiarization with ocular calibration is essential. It can help decrease anxiety, improve comfort, and increase cooperation during calibration, improving the accuracy of the data collected. The first task involves touching three diamond-shaped objects with the hand Figure 2a. The second task involves grabbing a coffee cup, moving it into a luminous circle, and then setting it down Figure 2b. The third task, Figure 2c involves enlarging the cup, and the fourth task involves reducing it Figure 2d. Finally, the fifth task, Figure 2e involves rotating a virtual object depicting a plant. These last four tasks require a grasping gesture (fine motor pinch movement between the thumb and index finger). Instructions are given orally, in written form, and via the appearance of a ghost hand demonstrating the expected gesture. Each task comprises three trials, and when three consecutive trials are successful, we consider the familiarization stage to be acquired, and we move on to the next task. 

We summarize all the steps of familiarization in Table 2, from the easiest level at the bottom to the most challenging level at the top.

### 2.3. Materials 

#### 2.3.1. HoloLens 2 

For this experiment, we use the MR headset HoloLens 2 [42], created by Microsoft, which displays an original tutorial developed by the company Actimage to facilitate the learning of grasping and manipulating holograms in MR. This application was developed using the Unity 3D MRTK game development engine and the Integrated Development Environment (IDE) Microsoft Visual Studio^®^ (2022). 

#### 2.3.2. Front Camera 

The HoloLens 2 is equipped with a front camera, located just above the visor, which allows visualization of gestures, holograms manipulated and the direction of the gaze. 

This latter, represented by a green sphere in Figure 3a, corresponds to the projection of the gaze in the foveal vision. This sphere is not visible to the participants. It is only visible on post-processed videos. Watching these videos allows us to measure the execution time for each trial and each participant. 

#### 2.3.3. Eye Tracking 

The HoloLens 2 headset has an application programming interface (API) to acquire a single gaze ray (origin and direction of gaze) at about 30 Hz. The predicted gaze is about 1.5 degrees in the visual angle around the target. We reused source code from the ARETT toolbox to retrieve the raw data from Microsoft’s Eye Tracking API for research [31]. The raw data obtained represents an x, y and z position of the gaze direction with a frequency of 30 Hz. To study the participants’ visual activity as a specific eye movement, we used the functions provided by the ARRET toolkit using the R programming language and its RStudio environment. All raw data were processed based on the study by Llanes-Jurado et al. [32] to categorize eye fixations in a 3D environment:An imputing function for not a number (Nan) values in the raw eye tracking data by using linear interpolation for temporal windows smaller than 75 ms;A function that classifies raw data as a fixation if the data within a 250 ms time window do not exceed a distance of 1.6 degrees;A function that merges fixations close to 1.6 degrees of distance within a 75 ms time window.

### 2.4. Outcome Measures

The familiarization success criteria are determined by the stages reached throughout the sessions. For evaluating the use of the tutorial, we focus on two dependent variables: the fixation duration (in milliseconds) on relevant cues (holograms in Figure 3a) and the execution time (in milliseconds) of each trial. The calculation of the fixation duration on relevant cues is based on the gaze fixation time within Areas of Interest (AOIs), defined around the holograms and instructions. AOIs are specified for each task involved and are dynamic, following any movements of the referent objects. Data processing is performed solely on fixation durations within the AOIs. The remainder of the perceptual field, the Mesh Figure 3b, accumulates all parts of the percept outside the AOIs.

Furthermore, we measure the execution time in milliseconds, calculated from the appearance of the task and instructions on the screen until the task is either completed or abandoned. This is done by reviewing the front camera footage from the HoloLens 2. This execution time allows us to assess the efficiency and speed with which participants manage to perform the various tasks proposed in the tutorial.

### 2.5. Study Procedure 

It is difficult to precisely determine the exact number of sessions during learning because it depends on the level of severity of autism, the nature of the learning as well as its design, such as the number of trials, the duration of the tests, the breaks etc.. In light of the studies cited in the familiarization portion of the introduction, the decision to conduct 16 sessions in this study strikes a balance between ensuring adequate familiarization and task learning for participants while considering the complexity of the design of the tasks, the different levels of autism severity of the participants and the technology interface used in the study. The familiarization procedure is individually proposed to people with ASD over 16 sessions, with one session per week lasting about 15 min. The sessions took place in the sports halls of the two partnering structures. If a participant has not reached stage 10 of familiarization within 16 sessions, the procedure is deemed unsuccessful. 

### 2.6. Results Analysis 

The results obtained from this study will enable the evaluation of the effectiveness of the familiarization procedure for people with ASD. The data analysis will focus on the participant’s progress over the sessions and the differences between fixation times and task 1 execution times for people with ASD compared to the control group.

Based on the results, some adjustments could be made to the familiarization procedure to make it even more effective and tailored to the specific needs of individuals with ASD. The ultimate goal of this study is to improve the learning tools and methods for individuals with ASD using MR technology, thereby promoting their inclusion and autonomy in various everyday activities.

### 2.7. Hypotheses and Statistical Treatments 

**Hypothesis 1** **(H1):**
*The average stage of familiarization for individuals with ASD will reach at least stage 10 after 16 sessions.*


**Hypothesis 2** **(H2):**
*Execution times and fixation durations differ between trials only during task 1 of the tutorial. This is analyzed using the Friedman non-parametric test on paired samples for the control group and the group with ASD. A Durbin-Conover post-hoc test will also be used to identify trials that show significant differences with a threshold of 5%.*


**Hypothesis 3** **(H3):**
*Execution times and fixation durations for each trial differ between groups only during task 1 of the tutorial. This is analyzed using the Wilcoxon non-parametric test on unpaired series between the control group and the group with ASD. A Durbin-Conover post-hoc test will also be used to identify trials that show significant differences with a threshold of 5%. All statistical analyses are performed using the R software and its RStudio environment. The figures are created using Excel software.*


## 3. Results 

### 3.1. Hypothesis 1: Familiarization 

On average, 81.81% of participants with ASD made three consecutive trials of the ‘Touch a Hologram’ task (stage 10) before the 16 scheduled sessions. Two of the 11 volunteers could not become familiarized with MR within the allotted sessions. On a group level with ASD, it takes an average of 6 sessions for them to reach stage 11 (Table of Figure 4). However, individual data shows that each person with ASD follows a different pace in familiarization progression (Figure 5). Only five volunteers managed to complete all stages. In parallel, all neurotypical individuals completed all tutorial tasks in a single session. 

Figure 4 shows the progression of each participant over the 16 scheduled sessions.

Statistical indicators reveal the following trends for the progression of participants with ASD throughout the 16 sessions: The median of the mean sessions: 11;Mean of the mean sessions: 11;Mean of the standard deviation sessions: 3.562430223.

It is important to note that participants with ASD showed a high degree of variability in their progression over the sessions (Figure 5), as evidenced by the high standard deviation. This variability highlights that each individual with ASD has a different familiarization pace and suggests that personalized approaches might facilitate their familiarization with MR.

### 3.2. Hypothesis 2: Intragroup Comparison

The non-parametric ANOVA test on paired samples was used to evaluate the intra-group difference of fixation times on the main hologram and execution times between the first three trials during the first task (Table 3).

Table 3 shows that during task 1, both variables differed between at least two trials accepting a minimum alpha risk of 5% in the control group. As for the group with ASD, there is a difference in task execution time 1 between at least two trials accepting at least a minimum alpha risk of 5%. We cannot conclude a difference in fixation durations on the main hologram between the trials of task 1 in the group with ASD.

We then used the Durbin Conover post-Hoc comparison test on paired series to compare trials between them on the fixation duration (Table 4) and execution times (Table 5). The tables refer to all the *p*-values obtained between the trials during task 1 according to the groups.

In the control group, the Durbin Conover tests indicate that the execution times and fixation durations are different between the first and second trial, as well as between the first and third trial, with a minimum alpha risk of 5%. However, we cannot conclude a difference in execution times and fixation durations between the second and third trial (*p*-value = 0.056 and *p*-value = 0.26) with a minimum alpha risk of 5%. In the group with ASD, the Durbin Conover tests show that we cannot conclude a significant difference in execution times and fixation durations between trials 1 and 2 and between trials 2 and 3, with a minimum alpha risk of 5%. However, a significant difference in execution times was observed between trials 1 and 3 (*p*-value = 0.026). As for the fixation duration, no significant difference was found between the trials. 

### 3.3. Hypothesis 3: Intergroup Comparison 

The non-parametric Wilcoxon comparison tests on unpaired series were used to evaluate the difference between groups of the first three trials during the first task, both for fixation durations on the main hologram and execution times.

Regarding the execution times (first row of Table 6 and Figure 6), the Wilcoxon tests indicate that we cannot conclude a significant difference between the two groups with an alpha risk of 5%. However, the Wilcoxon tests show that the fixation durations between the two groups present significant differences during the first and third trials with a minimum alpha risk of 5% (Figure 6). Thus, it is possible to observe that participants with ASD have different fixation durations than neurotypical participants during these trials.

## 4. Discussion

### 4.1. Familiarization

In our study, 81.81% of participants reached step 10 in the process of becoming familiar with MR (Figure 4). We observed overall progress despite difficulties encountered at steps 8, 9, and 11. Furthermore, individual variability indicates that each individual has different needs during familiarization, highlighting the importance of the initial steps (Figure 5). Step 9, corresponding to eye calibration, took an average of two sessions [43]. Attentional disorders among individuals with ASD disrupt this step, requiring sustained visual attention. Steps 11 to 14, which involve fine motor skills, also presented challenges for some participants.

On an individual scale, participants with normal motor skills easily passed the familiarization steps [44]. Among these, one took longer due to attention deficits [5]. Two others did not manage to surpass step 11 due to catatonia. This underscores the importance of a personalized approach to familiarization with MR, considering the specific challenges faced by a person with autism. Finally, we noted certain limitations of the MR used, such as the absence of haptic feedback [45,46], which played a role in the difficulty of performing certain tasks. The assumption that holograms can interact in some aspects, like real objects, was not fully learned. This is related to the difficulty people with ASD have in engaging in pretend play [47]. Future versions of HoloLens must include a haptic function to increase immersion [48]. Another volunteer took longer to reach step 10, partly due to attention deficit and sensitive, emotional regulation, which reduced session times [49].

Conversely, we stopped the study for two volunteers. One of them had a too strong sensitivity to wearing the headset. Despite positive reinforcements and considering his restricted interests [50], the volunteer did not progress and even regressed over time (yellow curve of Figure 5), characterized by refusals to wear the dummy headset [51]. The familiarization did not work. For the second volunteer (green curve of Figure 5), the familiarization to wearing the headset worked, as he managed to walk around with the real headset for several minutes. Our findings highlight the need for a personalized approach to familiarizing individuals with autism with mixed reality technology. This research contributes to a better understanding of how MR can effectively treat a person with autism [52].

In this part of the discussion, we analyze the effect of practice on the performance of the control group and participants with ASD. For NT participants, our results show that task repetition affected the two variables studied (execution time and fixation durations). Durbin-Conover tests show that execution times and fixation durations are always higher on the first trial than on the following two. This suggests that the speed of execution increased and reached a plateau from the second trial, which corroborates the idea that the data obtained follow a learning curve that adheres to a power law [53,54]. For participants with ASD, our results show how practice acted from the third trial affects execution time. This suggests that the group with ASD requires an additional trial to accomplish the task before execution time decreases.

We did not observe any effect of practice on the duration of fixation on the main hologram during the trials for the group with ASD, in contrast to the neurotypical group. This non-observation of the effect of practice within the group is likely because the first three trials of individuals with ASD are not necessarily successful the first time, unlike neurotypical individuals who completed task #1 in a single session. In designing learning, it is not mandatory to allocate time, but it can be helpful to observe whether the task is correct or false. Learning corresponds to a greater number of correct responses than the total number of trials over time. We did not implement this in our protocol because we wanted the volunteers to take as much time as needed, and it was difficult to estimate the maximum time for a trial as it can be easy for a neurotypical person but completely different for a person with ASD. Moreover, our experiments show that the maximum number of trials was reached mostly by neurotypical individuals (max(elapsed_time_1_) = 97,440 ms, max(elapsed_time_3_) = 42,020 ms), but the maximum for the second trial was a volunteer with ASD (max(elapsed_time_3_) = 117,080 ms). This explains the great variability in execution times during trial two concerning the group with ASD.

The skill, therefore, was not learned, and this did not influence the effect of practice on gaze. This observation aligns with studies showing that individuals with ASD generally need more time and repetitions to acquire new skills than neurotypical individuals [55,56]. This is linked to the difference we observed between the two groups and will detail in the following section. It is important to note that the sample size (9 people) is small and may also impact the results. Future studies with larger samples could help to clarify these observations further. When designing learning programs for individuals with ASD, it is essential to consider these differences in learning speed and adjust protocols accordingly. For example, providing more time and support for familiarization with the technology and task completion might be helpful. Furthermore, limiting the number of virtual objects and optimizing their layout in the user’s visual field may help improve visual focus and learning efficiency.

### 4.2. Group Comparison 

Regarding Hypothesis 3, our analysis focused on the intergroup comparison of execution times and fixation durations on the main hologram between the control group and the group with ASD. The results regarding execution times (first line of Table 6) do not show a significant difference between the two groups with a 5% alpha risk (Figure 6). This lack of difference may suggest that participants from both groups could perform the task within similar time frames, despite the specific challenges faced by individuals with ASD [57]. However, the first three trials of individuals with ASD were not necessarily successful since some could remove the headset or wander around before realizing the presence of the holograms. It is, therefore, essential to note that execution times only provide a partial measure of performance and that other factors, such as the quality and efficiency of interactions with holograms, must be considered for a complete evaluation of differences between groups. The non-difference is also partially explained by the sample size and great variability of the ASD group on the second trial, as we saw in the previous section (σ_trialNT #2_ = 6115.5 ms, σ_trialASD #2_ = 45,473.13 ms).

On the other hand, Wilcoxon tests show significant differences between the two groups (Figure 7) regarding fixation durations on the main hologram during the first and third trials (minimum alpha risk of 5%). These results suggest that individuals with ASD may have differences in how they direct their visual attention and interact with virtual stimuli compared to neurotypical participants [37]. Specifically, individuals with ASD might need more time to adapt to the virtual environment and holographic stimuli, which could explain the shorter fixation durations observed during the first and third trials.

We have highlighted a new indicator for learning in individuals with ASD and observed that participants with ASD had difficulty touching the hologram during the first three trials, which could be related to the characteristic attention deficits of this disorder [1]. However, focusing on the first three trials completed consecutively by the participants with ASD, we found that their execution speed and fixation durations were comparable to those of the control group (Figure 8). Indeed, Wilcoxon’s non-parametric comparisons on non-paired series showed that we could not conclude a difference between the two groups on execution speed (*p*-value _trial 1_ = 0.5748, *p*-value _trial 2_ = 0.9856, *p*-value _trial 3_ = 0.298), and this is also the case for fixation durations (*p*-value _trial 1_ = 0.2618, *p*-value _trial 2_ = 0.4916, *p*-value _trial 3_ = 0.664). These results suggest that learning through practice can modify the visual attention of individuals with ASD, enabling them to perform the task with performances similar to those of the control group. This observation aligns with previous studies showing that learning through practice can improve the performance of individuals with ASD in various tasks. The results of this study could help therapists to adapt and personalize the number of sessions needed to meet the needs and skills of individuals with ASD. However, in the group with ASD, Friedman’s tests on paired series indicate that the repetition of trials does not influence execution time (*p*-value = 0.6412) or fixation durations (*p*-value = 0.2231). These results do not show an influence of trial repetition on execution time and fixation durations. This lack of learning effect could be due to too small a sample of participants or the fact that participants with ASD had to perform several sessions before completing three consecutive trials. The learning effect may be less pronounced in individuals with ASD than in NTs, with a slower decrease in execution times and fixation durations.

To further investigate learning over time, we modeled execution speeds based on a decreasing exponential function in accordance with the literature on learning modeling [36,58]. The results show that the execution speeds of the two groups follow decreasing exponential laws with high coefficients of determination, indicating a gradual decrease in execution over the trials until a smooth plateau when the three trials are completed consecutively. The execution speeds of the control group follow an exponential law according to the equation y = 42272e^−0.615x^ with a coefficient of determination R^2^ = 0.9152. For the group with ASD concerning the first three trials, we obtain y = 74463e^−0.588x^ and a coefficient of determination of R^2^ = 0.5237, and the first three consecutive successful trials are modeled according to the equation y = 37898e^−0.395x^ and a coefficient of determination of R^2^ = 0.9533.

Results presented before having essential implications for therapists working with individuals with ASD, as they highlight the need to identify learning factors and adapt interventions accordingly. Several studies have emphasized the importance of an individualized approach to support learning in individuals with ASD [41]. By identifying the learning factors specific to each individual, therapists can better adapt their interventions and provide targeted support to improve the performance and skills of individuals with ASD. For example, therapists could use differentiated teaching strategies to meet the individual needs of individuals with ASD [59].

## 5. Conclusions

This study demonstrates that our familiarization method allowed 81.81% of our volunteers with autism to perform basic movements with the HoloLens 2. This study also shows the possibility of obtaining learning indicators using the HoloLens 2 headset. It highlights the ability of the HoloLens 2 to provide relevant learning indicators when using MR. The methodology used made it possible to detect changes in the participants' gaze, reflecting the acquisition of new skills throughout the experience. Although this data is not directly related to the MR itself, it is made possible thanks to the advanced functionalities of the sensors developed by Microsoft, such as eye tracking, voice recognition and recognition of facial emotions (Azure Face API), which are complementary to the MR.

From a research perspective, we plan to provide game-based apps to help people with autism improve their social skills using the HoloLens 2. The metrics discussed in this article will help evaluate these apps. We also recommend exploring potential enhancements to this technology, such as adding haptic features, increasing the acquisition frequency of built-in eye tracking, and using more detailed hand motor markers. Additionally, we are considering improving the tutorial by providing an avatar as a standalone assistant to help the user complete the tutorial in co-op.

There are limitations to our current study that could be overcome in future research. For example, we will need to spend more time familiarizing participants with the headset and increasing the sample size. In addition, we plan to work with software engineers to develop new user interfaces and analysis tools, such as machine learning or artificial intelligence, to offer personalized and autonomous learning.

In conclusion, this research highlights the importance of using innovative technologies, such as the HoloLens 2 headset, to facilitate the analysis and understanding of learning mechanisms. The results offer new perspectives on the potential applications of these tools in social cognition and associated therapies, particularly for people with ASD. Future work could extend these approaches to other patient populations and explore the implementation of tailored therapies using mixed reality. This could not only open new therapeutic avenues for people with ASD but also for other people with neurodevelopmental disorders or disabilities. Therefore, the future of research in this field is very promising and offers many opportunities to improve therapeutic interventions through MR technology.

## Figures and Tables

**Figure 1 sensors-23-06304-f001:**
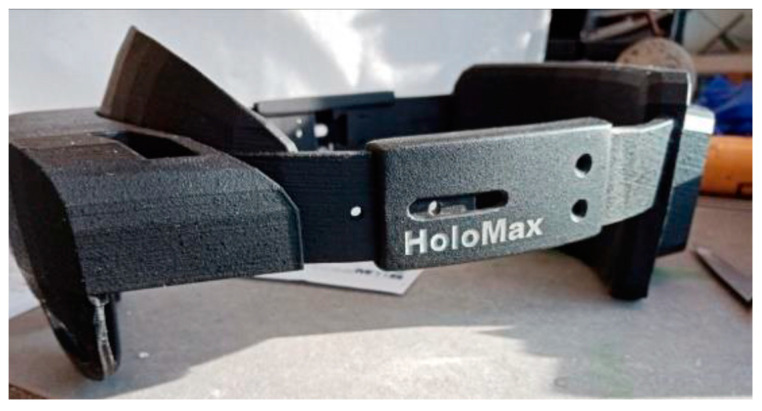
Overview of the dummy HoloMax headset.

**Figure 2 sensors-23-06304-f002:**
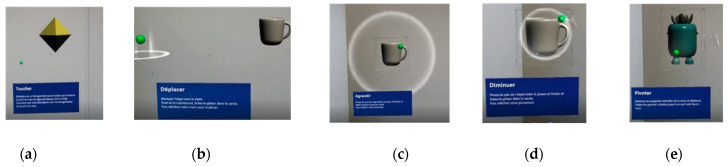
Screenshots demonstrating a sequence of tutorial tasks during a volunteer’s session. The sequence progresses from left to right, as labeled (**a**–**e**).

**Figure 3 sensors-23-06304-f003:**
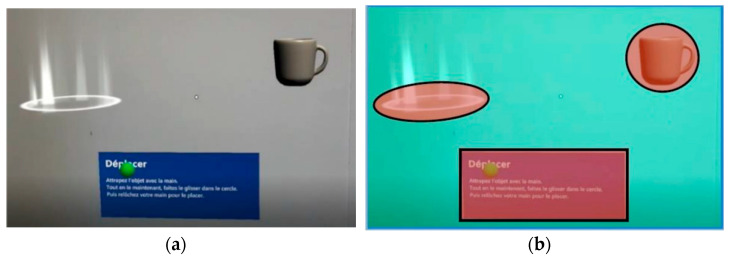
Screenshot of the second task (**a**) during a session and its semantic representation (**b**) where the AOIs are highlighted in red and the background in green.

**Figure 4 sensors-23-06304-f004:**
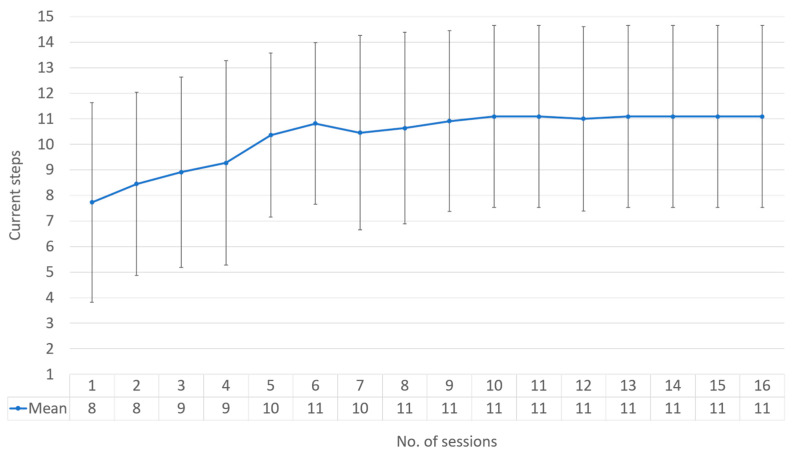
Average progression of successful HoloLens 2 familiarization stages by session.

**Figure 5 sensors-23-06304-f005:**
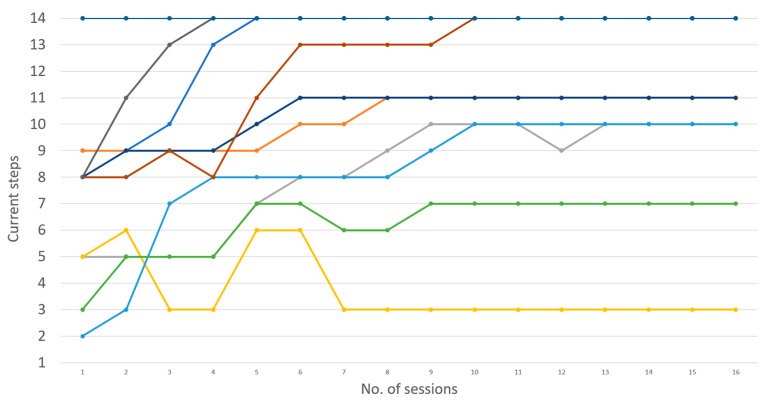
Individual progressions of the HoloLens 2 familiarization stages according to the sessions.

**Figure 6 sensors-23-06304-f006:**
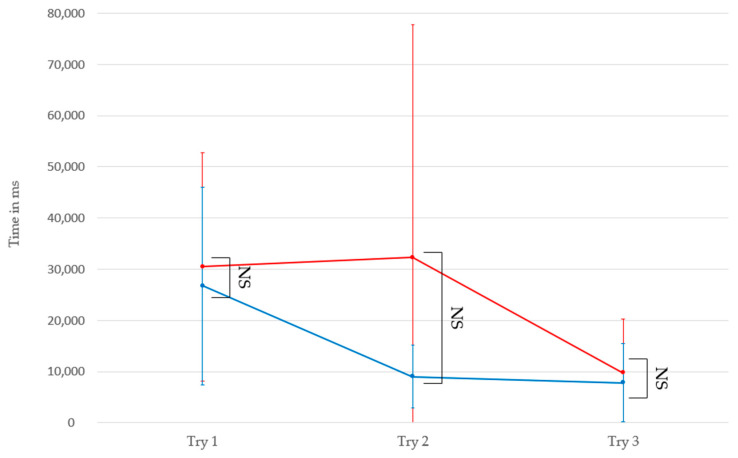
Intergroup comparison of execution times based on the first three trials during Task 1. The blue curve represents the average execution times of the control group, and the red curve represents the average for the group with ASD. NS: Not significant.

**Figure 7 sensors-23-06304-f007:**
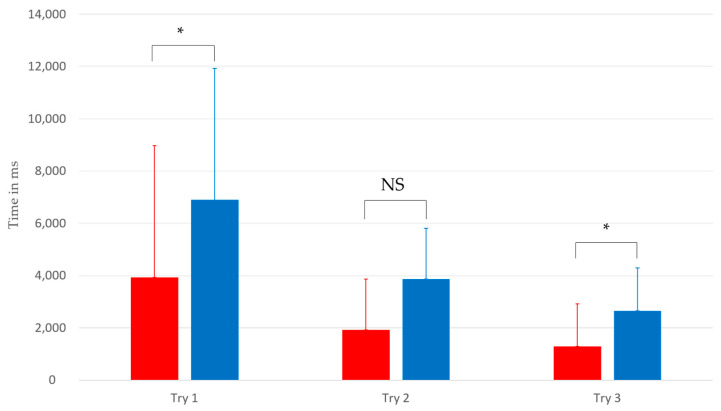
Intergroup comparison of fixation durations on the hologram based on the first three trials during Task 1. The blue bands represent the average fixation times of the control group, and the red bands represent the average for the group with ASD. *: *p*-value less than 0.05. NS: Not significant.

**Figure 8 sensors-23-06304-f008:**
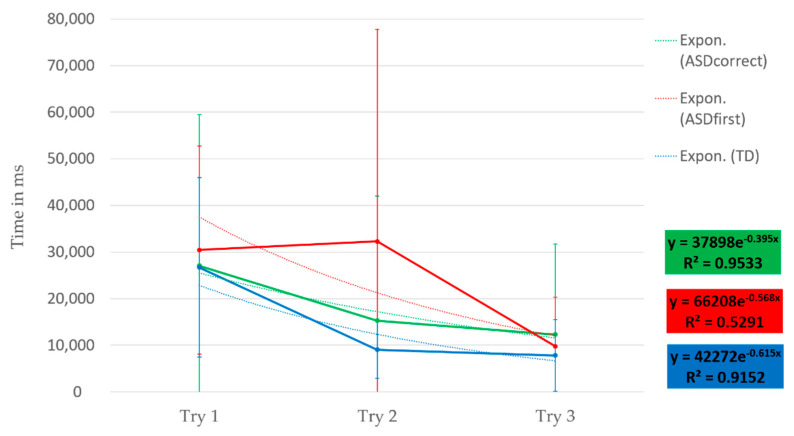
Execution times as a function of the first three trials of the control group (blue curve), the group with ASD (red curve), and the first three consecutive successful trials (green curve) during task 1. The dotted curves represent the exponential curves of execution times.

**Table 1 sensors-23-06304-t001:** Census of sociodemographic data of individuals with ASD.

ID	Age (Years)	Gender	Vineland-II (Standard Note Sum)	Diagnosis
001PR	25	Male	30	ASD
002FV	22	Male	20	ASD
003FM	28	Female	20	ASD
004TM	19	Male	37	PDD * with language disorder
005RK	19	Male	56	PDD * atypical disorder
006TC	28	Male	20	ASD
007HC	25	Male	20	ASD
008ST	14	Male	20	ASD
009MA	15	Male	23	ASD
010DF	30	Male	20	ASD
011BN	40	Female	20	ASD

* Pervasive Developmental Disorder.

**Table 2 sensors-23-06304-t002:** The increasing steps (from top to bottom) of familiarization with wearing the headset and then with mixed reality.

Type of Familiarization	Steps
Headset Familiarization	1. Headset Presentation
	2. Touch HoloMax
	3. Wear HoloMax for a few seconds
	4. Wear HoloMax for a few minutes
	5. Move around with HoloMax
	6. Move around with the HoloLens 2
Mixed Reality Familiarization	7. View holograms
	8. Interact with holograms
	9. Eye Tracking Calibration
	10. Touch in the tutorial
	11. Move to the tutorial
	12. Enlarge in the tutorial
	13. Shrink in the tutorial
	14. Pivot in the tutorial

**Table 3 sensors-23-06304-t003:** Census of all *p*-values obtained during the non-parametric ANOVA test on paired samples according to the two dependent variables during the first task.

Group	*p* Value for Differences in Fixation Duration on the Main Hologram during Task 1	*p* Value Regarding Runtime Differences on the Main Hologram during Task 1
Control	1.069 × 10^−13^	1.587 × 10^−11^
With ASD	0.131	0.03192

**Table 4 sensors-23-06304-t004:** Census of *p* values obtained during Durbin Conover’s post-Hoc comparison on paired series of fixation duration between trials and according to each task.

Execution Times	Try n°1 vs. n°2	Try n°1 vs. n°3	Try n°2 vs. n°3
Control group	*p* value = 1.9 × 10^−5^	*p* value = 2.9 × 10^−11^	*p* value = 0.056
Group with ASD	*p* value = 0.613	*p* value = 0.026	*p* value = 0.225

**Table 5 sensors-23-06304-t005:** Census of the *p*-values obtained during the Durbin Conover post-Hoc comparison on paired series of execution times between trials and according to each task.

Fixation Duration	Try n°1 vs. n°2	Try n°1 vs. n°3	Try n°2 vs. n°3
Control group	*p* value = 3.4 × 10^−8^	*p* value = 1.1 × 10^−12^	*p* value = 0.26
Group with ASD	*p* value = 0.27	*p* value = 0.13	*p* value = 0.92

**Table 6 sensors-23-06304-t006:** Census of *p*-values obtained during the non-parametric Wilcoxon comparison on unpaired series of fixation duration between trials and according to each task.

NT vs. TSA	Try n°1	Try n°2	Try n°3
Execution times	*p* value = 0.6251	*p* value = 0.4691	*p* value = 0.2505
Fixation duration	*p* value = 0.03422	*p* value = 0.1112	*p* value = 0.01384

## Data Availability

The data presented in this study are available on request from the corresponding author. The data are not publicly available due to data privacy.

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
