# Peer review of "Familiarization with Mixed Reality for Individuals with Autism Spectrum Disorder: An Eye Tracking Study"

_sensors, 2023, doi:10.3390/s23146304_

Round 1
Reviewer 1 Report
After reading this article, I acknowledge your work has potential application value. But there are still some shortcomings could be improved.
1. The five images in Figure 2 are misaligned. The distance between image A and picture B is too large, and the top and bottom of picture E are obviously not aligned. (section 2.2.2)
2. The legend and image body of Figure 2 are paginated. (section 2.2.2)
3. Too much white space in page ten and page twelve.
4. The method section did not provide specific formulas for calculating the results, and the experimental data lacked rigorous theoretical support.
5. There are too few comparative experiments, and there is no comparison with the latest methods in this direction, and it is not convincing enough.
6. The discussion section is too redundant, does not repeat the results given in verbatim or in detail, and describes the important or unique contribution of the work objectively and fairly.
7. The experimental part is not detailed enough, and the conclusions drawn may not be sufficient to support your theory.
8. The conclusion suggests summarizing the shortcomings of one's own research and providing future recommendations. (Conclusions)

Author Response
Dear Reviewer,
Thank you for your detailed and constructive comments. We have made the following changes in response to your suggestions:
1. We rearranged Figure 2 to ensure perfect alignment and added a white box to improve clarity.
2. We also adjusted the layout to ensure that the caption and the body of Figure 2 are on the same page.
3. Excessive white space on pages 10 and 12 has been removed for better space utilization.
4. Regarding the need for specific formulas to calculate results, we have provided references to key studies that support our eye tracking methodology, allowing interested readers to dig deeper into this topic. Furthermore, the inclusion of the explanatory equations of the functions used in our study, which come from the work of the study [34], could significantly lengthen the text, potentially by three pages or more, which would divert the reader's attention from the heart of our work. In addition, we have provided the equations of the execution speed trends according to the three tests (figure 8 and section 4.2).
5. You mentioned the lack of experimental comparisons. It is true that our methodology is unique, combining different fields, which makes direct comparison with other studies difficult. However, we have highlighted points of agreement between our results and those of previous studies that address similar areas by adding the relevant literature in section 4.2.
6. Regarding discussion section redundancy, we have revised and rewritten this section to be more concise and to the point.
7. We have also adjusted our conclusion to strengthen the links between our experimental results and our theoretical claims.
8. Finally, we have added the limitations of our study as well as future research perspectives to the conclusion, as per your recommendation. We hope these changes address your concerns and improve the quality of our work. We look forward to any further comments you may have.
You will find all the changes in the attached document.

Reviewer 2 Report
In this study, entitled “Familiarization with Mixed Reality for Individuals with Autism Spectrum Disorder: An Eye Tracking Study”, the authors investigate a method for familiarizing eleven individuals with severe ASD with the HoloLens 2 headset and the use of MR technology through a tutorial.
I consider the topic original, and the study has the potentiality of being shared with the scientific community.
The manuscript is very well written; clear, precise, and easy to understand. It adds new knowledge and perspectives in Autism Spectrum Disorders (ASD) through the use of technology. It offers new perspectives on skill acquisition indicators useful for supporting neurodevelopmental disorders and contributes to a better understanding of the neural mechanisms underlying learning in Mixed Reality for individuals with ASD.
the references are appropriate.
Although the study has the potentiality of being shared with the scientific community, I believe that the manuscript would benefit from a minor revision with the attempt to better support their experimental setting.
1. Methods section:
- More information should be provided about the participants’ characteristics.
2. The Discussion should be enriched with the existing theory. The authors should clearly describe the scientific evidence that supports their findings. In addition, they should start with a first paragraph describing the main aims and then the main results.
Kind regards
-
Author Response
Dear Reviewer,
Thank you for your detailed and constructive comments. We have made the following changes in response to your suggestions:
- I added more information about participants in table 1 of section 2.2.1.
- I modified the discussion part according to your request.
You will find all the changes in the attached document.
Regards.

Round 2
Reviewer 1 Report
All my comments and suggestions are addressed in revised version.